# Impact of Maternal Fish Consumption on Serum Docosahexaenoic Acid (DHA) Levels in Breastfed Infants: A Cross-Sectional Study of a Randomized Clinical Trial in Japan

**DOI:** 10.3390/nu15204338

**Published:** 2023-10-11

**Authors:** Ayu Kasamatsu, Hiroshi Tachimoto, Mitsuyoshi Urashima

**Affiliations:** 1Division of Molecular Epidemiology, The Jikei University School of Medicine, 3-25-8 Nishi-Shimbashi, Minato-Ku, Tokyo 105-8461, Japan; ayu.kasamatsu@gmail.com; 2Department of Pediatrics, The Jikei University School of Medicine, 3-25-8 Nishi-Shimbashi, Minato-Ku, Tokyo 105-8461, Japan; hiroshitachimoto@gmail.com

**Keywords:** infants, Japanese, seafood, breast feeding, docosahexaenoic acid, eicosapentaenoic acid, fatty acids, milk, omega-3, omega-6

## Abstract

Docosahexaenoic acid (DHA), an essential n-3 long-chain polyunsaturated fatty acid (LCPUFA) abundant in fish, is crucial for infant brain development. We investigated the associations between maternal dietary habits, infant feeding patterns, and serum levels of DHA and other LCPUFAs in infants aged 5–6 months in Japan, where fish consumption is high. This cross-sectional study used serum samples from 268 infants enrolled in a randomized clinical trial. The frequency of mothers’ consumption of 38 food items and infant feeding patterns were prospectively surveyed. Cow’s milk formula (CMF) supplemented with 15.9% linolenic acid, 1.6% α-linolenic acid, 0.40% DHA, and 0.27% arachidonic acid was used. Significant positive associations with infants’ serum DHA levels were found for “Blue-back fish” (rho = 0.24; *p* = 0.0001) and “White fish” (rho = 0.25, *p* = 0.0001). The combined variable “Blue-White fish” was found to be significantly associated with higher serum DHA levels in infants (rho = 0.29, *p* < 0.0001). Predominantly breastfed infants had significantly higher serum DHA levels than those fed more CMF (rho = 0.32, *p* < 0.0001). After multivariate analysis, “Blue-White fish” and “Feeding patterns” remained significantly and independently associated with serum DHA levels. These findings suggest that frequent consumption of “Blue-back fish” and/or “White fish” by lactating mothers, along with prioritizing breastfeeding over DHA-supplemented CMF, might effectively increase infants’ serum DHA levels.

## 1. Introduction

Exclusive breastfeeding is recommended for the first 6 months after birth [1,2] due to its numerous benefits for infants [3], one of which is the potential to enhance intelligence via unknown mechanisms [3,4,5]. Breastfed infants exhibit higher plasma levels of docosahexaenoic acid (DHA) than those fed DHA-unsupplemented infant formula [6,7,8]. DHA is an n-3 fatty acid belonging to the group of long-chain polyunsaturated fatty acids (LCPUFAs) and is essential for infant brain development [9], aiding myelination and improving the speed of electrical impulses in neurons [10]. Indeed, higher serum DHA levels are linked to increased brain volumes in neonates [11]. The process of incorporating DHA into the brain membrane structure in early life relies on its maternal transfer and dietary intake, as well as endogenous LCPUFA synthesis, as there is no de novo LCPUFA synthesis [12]. Thus, efforts have been made to maintain the DHA levels of formula-fed infants at levels comparable to those of breastfed infants using DHA supplementation. Recently, evidence of the clinical benefits of DHA supplementation has been accumulating. Multiple randomized controlled trials (RCTs) have suggested the possible effect of DHA supplementation on the development of visual acuity and cognitive outcomes in both preterm and full-term infants, although they have shown inconsistent results due to variations in outcome measurements and in DHA supplementation dosages among studies [13,14,15]. Some RCTs have also shown improvements in cognitive outcomes in very low birth weight infants with DHA supplementation [16,17]. Furthermore, DHA supplementation not only offers short-term advantages but also long-term benefits, enhancing brain structure and function and improving neurochemical outcomes in children aged 5 years and older [16,18,19].

To promote such favorable effects, it is recommended that lactating mothers consume seafood, which is rich in DHA [20]. This recommendation is based on the evidence that maternal dietary choices can affect breast milk fatty acid composition [21]. However, it remains unknown whether lactating mothers’ diets affect their infants’ fatty acid levels. Moreover, blood DHA levels were reported to be equivalent between breastfed infants and LCPUFA-supplemented formula-fed infants during the first several months of life in previous studies, although those studies were mostly from Western countries [22,23,24,25].

Japanese dietary habits are characterized by the consumption of considerable amounts of seafood. In fact, Japanese people consume more fish, with an average intake of 31.2–52.5 g/day, than Caucasian Americans, who consume 8.9 g/day [26]. The high consumption of fish in Japan makes it an ideal location to examine the effects of maternal fish consumption on serum DHA levels in breastfed infants. Thus, this study aimed to investigate the association between the dietary habits of lactating mothers and serum DHA and other LCPUFA levels in Japanese term infants before they start eating solid foods. We hypothesized that, in contrast to studies from Western countries, Japanese breastfed infants might have higher serum DHA levels than those fed cow’s milk formula (CMF) supplemented with LCPUFAs. Hence, this study also aimed to elucidate the association between infant feeding patterns and their serum DHA levels.

## 2. Materials and Methods

### 2.1. Study Design

This study was designed as a pre-specified cross-sectional study conducted as a supplemental analysis of the Atopy Induced by Breastfeeding or Cow’s Milk Formula (ABC) trial [27]. Since the effects of allergen avoidance in neonates on reducing the risk of food allergy are not well established, the trial aimed to determine if avoiding or introducing CMF for at least the first three days of life might decrease the risk of sensitization to cow’s milk protein and clinical food allergies. Details of the background and methods of the ABC trial are described elsewhere [27]. Briefly, in the ABC trial, newborn infants were randomized to breastfeeding (BF) with or without amino acid–based elemental formula (EF) for at least the first 3 days of life group (BF/EF group) or a BF supplemented with CMF (≥5 mL/day) from the first day of life to 5 months of age group (BF + CMF group) and were followed up until their second birthday. Enrollment began on 1 October 2013, and follow-up was completed on 31 May 2018 at a single university hospital in Tokyo, Japan. Written, informed consent was obtained from the parents of all the enrolled infants. The trial protocol was approved by the ethics committee of Jikei University School of Medicine and the institutional review board of Jikei University Hospital (25-057(7192)). The trial was registered with the UMIN Clinical Trials Registry (UMIN000011577). This cross-sectional study used information from a maternal dietary questionnaire and serum fatty acid levels in blood samples from infants aged 5 months that were specifically collected for this trial in order to analyze their association at this single time point.

### 2.2. Study Population

The ABC trial included infants at risk of atopy due to at least one of their parents or siblings having current and/or past atopic diseases (e.g., asthma). Infants whose parents intended to exclusively provide breastfeeding or CMF before birth, or infants who were born at less than 36 weeks’ gestational age, had a birth weight of less than 2000 g, or had serious congenital anomalies (e.g., cleft palate) were excluded.

### 2.3. Infant Formula and Intervention

To evaluate sensitization to cow’s milk protein, this trial incorporated both CMF (Meiji Hohoemi^®^, Meiji Holdings Co., Ltd., Tokyo, Japan) and EF (Meiji Elemental^®^ formula, Meiji Holdings Co., Ltd.) alongside breast milk; EF does not contain cow’s milk protein, but it does contain fundamental nutrients in approximately equivalent proportions to CMF. The CMF was supplemented with α-linolenic acid (ALA) = 1.6%, DHA = 0.40%, linolenic acid (LA) = 15.9%, and arachidonic acid (AA) = 0.27% per total fatty acid weight, but not with eicosapentaenoic acid (EPA). The EF was supplemented with ALA = 12.4% and LA = 60.2%. Details of the fatty acid composition and other nutritional components of the infant formulas are provided in the Appendix A. Newborns were centrally assigned in a 1:1 ratio to the BF/EF and BF + CMF groups using permuted blocks of 4 by computer randomization. Infants in the BF/EF group were to avoid CMF for at least the first 3 days of life but were allowed to receive EF when mothers believed that the amount of breast milk was not enough. If they added more than 150 mL/day of EF to breastfeeding for three consecutive days, the EF was switched to CMF after the fourth day. Infants in the BF + CMF group were to receive at least 5 mL/day of CMF from the first day of life and at least 40 mL/day after 1 month of age until the first blood test at 5–6 months of age or before starting solid foods to supplement breastfeeding.

### 2.4. Sample Size

The primary outcome of the ABC trial was defined as a CM-IgE level of 0.35 UA/mL or greater at 24 months of age. For sample size calculation, it was assumed that this outcome would be achieved in 10% of one group and 25% of the other, with a bilateral type I error of 5%, 90% power, and an estimated 3% loss to follow-up. Therefore, we calculated that a sample size of 300 participants in a 1:1 ratio would be required to detect this difference.

### 2.5. Data Collection

Data on maternal age, maternal body mass index, gestational weeks, birth weight, and neonatal sex were obtained from the neonatal summary record at birth. Participating parents were prospectively interviewed regarding their child’s daily amount of CMF intake during follow-up visits to the outpatient clinic when their child was 5–6 months old. The feeding patterns were prospectively classified into the following four grades based on the pre-specified cutoff points of CMF intake: BF with CMF > 100 mL/day, BF with CMF 40–100 mL/day, BF with CMF 5–40 mL/day, and exclusive BF. If the feeding patterns were not recorded at 5–6 months of age, data on their feeding patterns at 3 months of age were used instead. Moreover, during a previous visit, mothers were given a self-administered questionnaire regarding their diet, which included 38 food items. The completed answers were to be brought to the visit when the child was 5–6 months old. In the questionnaire, mothers were asked to check the most applicable box on a 6-point scale, ranging from “rarely” to “three times a day”, for the frequency of consumption of each listed food item over the previous month.

### 2.6. Serum Fatty Acid Measurements and Primary/Secondary Outcomes

Blood was collected from the infants at 5–6 months of age, prior to the introduction of solid foods. In the ABC trial, 25-hydroxyvitamin D, total IgE, and antigen-specific IgE levels were measured. If sufficient residual serum samples were available, they were utilized to measure the composition of 24 serum fatty acids. The serum samples were stored at −80 °C and sent to Standard Reference Laboratory Inc. (Tokyo, Japan) for analysis. The concentration of each fatty acid was measured by gas chromatography-mass spectrometry using a calibration curve method and expressed as the weight percentage of total fatty acids [28]. The primary outcome was set as the infant serum DHA level. The secondary outcomes were infant serum EPA, AA, ALA, and LA levels. Moreover, n-3 LCPUFA, which includes the sum of ALA, EPA, docosapentaenoic acid (DPA), and DHA, as well as n-6 LCPUFA, which includes the sum of LA, γ-linolenic acid, eicosadienoic acid, dihomo-γ-linolenic acid (DGLA), AA, and docosatetraenoic acid (DTA), were also evaluated as secondary outcomes.

### 2.7. Statistical Analysis

Spearman’s rank correlation (rho) was employed to assess the associations between the frequency of maternal food intake of the 38 listed items or grades of feeding patterns and seven outcomes, and among n-3 LCPUFAs and n-6 LCPUFAs to quantify the strength of correlations a rho value of ≥0.9 was considered a very strong correlation, 0.9 > rho ≥ 0.7 was strong, 0.7 > rho ≥ 0.4 was moderate, 0.4 > rho ≥ 0.1 was weak, and rho < 0.1 was a negligible correlation [29]. Univariate and multivariate regression analyses were also conducted to explore factors associated with serum DHA levels in infants. The Bonferroni correction was applied to account for multiple comparisons (e.g., 300 times), and a two-sided *p* value of <0.00016 was considered statistically significant. All data were analyzed using Stata, version 17.0 (StataCorp, College Station, TX, USA).

## 3. Results

### 3.1. Study Population

The ABC trial included 312 pregnant women who were randomly assigned to either the BF/EF group or the BF + CMF group from the first day of their infant’s life in a 1:1 ratio (Figure 1); there was no loss to follow-up until 5–6 months of age. Blood samples were collected from 309 out of 312 infants at 5–6 months of age. A total of 268 residual serum samples from the infants were available for measurement of the 24 different types of fatty acids (Cohort I). In addition, food intake frequency questionnaires were prospectively collected from 254 participants (Cohort II), and 223 participants were prospectively interviewed regarding their infant’s feeding patterns (Cohort III). The number of participants in each grade of feeding patterns was as follows: (1) CMF >100 mL/day (*n* = 121); (2) CMF 40–100 mL/ day (*n* = 10); (3) CMF 5–40 mL/day (*n* = 23); and (4) exclusive BF (*n* = 69). None of the infants were receiving EF at the time of blood sampling. The participants’ characteristics are shown in Table 1.

### 3.2. Serum Fatty Acid Compositions at 5–6 Months of Age

The present study first investigated fatty acid compositions in 268 infants (Cohort I), analyzing 24 different types of fatty acids (Table 2). For n-3 LCPUFA, strong positive associations were observed between serum levels of EPA, DPA, and DHA, while a moderate negative association was found between ALA and DHA (Figure 2A). Additionally, moderate associations were found between DGLA, AA, and DTA levels among the n-6 LCPUFAs, but negligible associations were found between LA and other fatty acids (Figure 2B). Interestingly, strong negative associations were observed between monounsaturated fatty acids (MUFA) and n-6 LCPUFA, and moderate negative associations were found between saturated fatty acids (SFA) and n-6 LCPUFA, as well as between MUFA and n-3 LCPUFA. Conversely, negligible associations were observed between n-3 and n-6 LCPUFA (Figure 2C).

### 3.3. Frequency of Maternal Intake of the Listed Items and Serum DHA Levels in Infants

This study next examined the association between the frequency of consumption of 38 food items by lactating mothers and serum DHA levels in their infants (Table 3). Of the 38 food items, only two types of fish, “Blue-back fish” (rho = 0.24, *p* = 0.0001) (Figure 3A) and “White fish” (rho = 0.25, *p* = 0.0001) (Figure 3B), were found to have significant positive associations with serum DHA levels in infants. In contrast, other types of fish, such as salmon, tuna, and swordfish, as well as food categories such as nuts, dairy products, eggs, vegetable oil, fried foods, meat, and beans, did not have any significant associations with DHA levels. A new variable, “Blue-White fish”, created by combining the frequency of consumption of these two fish types, was found to be significantly associated with higher levels of serum DHA in infants (rho = 0.29, *p* < 0.0001) (Figure 3C). When stratified by infant sex, the relationship between frequency of “Blue-White fish” intake and serum DHA levels persisted in both males (rho = 0.40, *p* < 0.0001) and females (rho = 0.39, *p* < 0.0001), showing no evident interaction of sex (*p* for interaction = 0.112) (Appendix A).

### 3.4. Frequency of Maternal Intake of the Listed Items and Serum EPA and Other LCPUFA Levels in Infants

The study also examined the associations between the frequency of maternal intake of certain foods and the levels of serum EPA and other LCPUFAs in their infants. “Blue-back fish” and “White fish”, which were found to be associated with DHA levels, also showed significant positive associations with EPA levels in the infants (rho = 0.32, *p* < 0.0001 for both) (Figure 4A,B). The combined intake of “Blue-White fish” was also significantly and positively associated with EPA levels (rho = 0.40, *p* < 0.0001) (Figure 4C). Weak negative and positive trends were observed between “Junk food” (rho = −0.19, *p* = 0.003) (Figure 4D) and “Beans” (rho = 0.17, *p* = 0.008) (Figure 4E), respectively, and EPA levels, although their *p* values did not reach statistical significance using the cutoff point of *p* = 0.00016.

Similarly, the intake of ‘Blue-back fish’ (rho = 0.29, *p* < 0.0001) (Figure 5A), “White fish” (rho = 0.31, *p* < 0.0001) (Figure 5B), and the combined intake of “Blue-White fish” (rho = 0.37, *p* < 0.0001) (Figure 5C) were significantly and positively associated with n-3 LCPUFA levels. Weak positive and negative trends were also observed between “Junk food” (rho = −0.16, *p* = 0.009) (Figure 5D) and “Beans” (rho = 0.18, *p* = 0.004) (Figure 5E) and n-3 LCPUFA levels, respectively. However, none of the 38 food items analyzed showed significant associations with serum levels of AA, LA, ALA, or n-6 LCPUFA.

### 3.5. Feeding Patterns and Serum DHA and Other n-3 LCPUFA Levels in Infants

Examination of the association between the infants’ feeding patterns and their serum n-3 LCPUFA levels revealed that infants who were predominantly breastfed had significantly higher levels of serum DHA (rho = 0.32, *p* < 0.0001) (Figure 6A), EPA (rho = 0.35, *p* < 0.0001) (Figure 6B), AA (rho = 0.30, *p* < 0.0001) (Figure 6C), and n-3 LCPUFA (rho = 0.36, *p* < 0.0001) (Figure 6D) compared to those who received larger amounts of CMF to supplement breastfeeding. Conversely, infants who were fed less CMF had significantly lower levels of serum LA (rho = −0.33, *p* < 0.0001) (Figure 6E) compared to those who were predominantly breastfed. No significant associations were found between feeding patterns and serum ALA (Figure 6F) or n-6 LCPUFA levels (Figure 6G).

### 3.6. Factors Associated with Serum DHA Levels in Infants: Univariate and Multivariate Regression Analyses

We assessed the infants’ serum DHA levels in relation to six variables: “Blue-White fish” consumption, feeding patterns, group allocation in the ABC trial, maternal age, maternal body mass index, gestational weeks, birth weight, and infant sex (Table 4). Univariate analysis revealed that both “Blue-White fish” consumption and “feeding patterns” were significantly associated with serum DHA levels, while there was no significant association for the other factors (Model I). In the subsequent multivariate analysis (Model II), including the two significant factors identified in univariate analyses, “Blue-White fish” consumption and “feeding patterns” remained significant variables. After adjusting for all six variables, both “Blue-White fish” consumption and “feeding patterns” remained significant variables (Model III). Notably, when infant sex was included in the univariate analysis for “Blue-White fish” and serum DHA levels, the coefficient for “Blue-White fish” remained virtually unchanged (coefficient = 0.29 for both), showing no evidence of a confounding effect of sex.

## 4. Discussion

To the best of our knowledge, this is the first study to explore the association between maternal diet and serum LCPUFA levels in infants who have not yet started solid foods. The results indicated that the infants of lactating mothers who frequently consumed “Blue-back fish” and/or “White fish” from the list of 38 food items had higher levels of serum DHA than infants whose mothers did not consume these fish regularly. The study also found that infants who were predominantly breastfed had higher levels of serum DHA compared to those who were formula-fed. Even following multivariate analysis, frequent intake of “Blue-White fish” by lactating mothers and “feeding patterns” remained significant factors related to infant DHA levels, indicating that both factors were independently associated with higher serum DHA levels in infants. These results suggest that higher serum DHA levels might be more effectively achieved by frequent maternal consumption of “Blue-back fish” and/or “White fish”, or by prioritizing breastfeeding rather than relying on CMF supplemented with DHA.

A previous meta-analysis revealed that the concentration of DHA in breast milk varies considerably by country and is generally higher in coastal populations with a higher consumption of seafood [30]. A large Canadian birth cohort study showed that maternal intake of fish oil supplements and cold-water fish (such as salmon, mackerel, and bluefish) was positively associated with DHA and n-3 LCPUFA levels in breast milk but not with AA and n-6 LCPUFA levels [31]. Similarly, the Japanese Human Milk Study reported that maternal consumption of grilled fish was associated with increased DHA levels in breast milk [32]. These previous reports support the finding of this study that serum DHA levels are higher in infants fed by mothers who frequently consume “Blue-White fish” as compared to those who only infrequently consume such fish. Hence, our findings imply that mothers can potentially enhance their infants’ DHA levels by frequently consuming “Blue-back fish” and/or “White fish,” even when breastfeeding is limited due to milk supply or maternal lifestyle. In Japan, these types of fish are affordable and readily available, often as convenient, ready-to-eat options, such as canned or convenience store offerings. This accessibility makes it feasible for even busy mothers to incorporate these fish into their daily diets. It should be cautioned, however, that causality was not determined in this study due to the nature of the study design. Additionally, the study did not assess LCPUFA levels in breast milk or maternal serum, necessitating further longitudinal investigations to comprehensively evaluate the relationship between maternal dietary intake of these fish, LCPUFA levels in maternal serum and breast milk, and infant DHA levels.

No interaction of sex was evident in the association between maternal “Blue-White fish” intake frequency and infant serum DHA levels in this study. As demonstrated in a systematic review predominantly involving adults, females have higher plasma DHA levels (%) compared to males. This could be attributed to differences in n-3 LCPUFA synthesis from ALA, dietary patterns, and age [33]. However, only limited data on children are available, with studies showing no sex differences in plasma or erythrocyte membrane LCPUFA composition among individuals from infancy to young adulthood [34]. These previous findings are consistent with our observations. On the other hand, DHA levels in breast milk are suggested to be higher when the infant is female [35]. Thus, further studies are warranted to elucidate the effects of sex on the association between maternal diet, LCPUFA levels in breast milk, and the levels in infant serum.

The present study also revealed that serum DHA levels were higher in breastfed infants compared to those fed CMF, even when it was supplemented with DHA. Contrary to our findings, most studies from Western countries indicated similar [22,23,24,25] or lower [36,37] blood DHA levels in 3–9-month-old, full-term, breastfed infants compared to those fed LCPUFA-supplemented formula. This discrepancy might be due to the comparable or lower DHA levels observed in breast milk versus formula in these studies. Conversely, Japanese mothers in previous studies exhibited higher DHA levels in breast milk (ranging from 0.53% to 1.10%) [38,39,40], exceeding the 0.4% DHA supplementation in CMF used in our study. Indeed, a previous Japanese study involving 6-month-old infants showed a similar trend in mean erythrocyte membrane DHA as that in our present findings [41]. Data from our Japanese study thus offer insights into the potential influence of relative DHA levels in breast milk and infant formula on the relationship between feeding patterns and infant blood DHA levels.

In addition to DHA, AA is also crucial for infant growth, brain development, and overall health [42]. Thus, 0.27% AA was added to the CMF used in this study. However, contrary to expectations, the results showed that breastfed infants had higher serum AA levels compared to those fed CMF, even when it was supplemented with AA. Interestingly, unlike DHA, maternal consumption of specific foods, such as fish, did not appear to affect serum AA levels in infants. AA is found in small amounts in a variety of animal-derived foods, such as meat, poultry, eggs, fish, and dairy products [43]. This contrasts with DHA and EPA, which are primarily found in seafood. As such, EPA and DHA were associated with the consumption of “Blue-back fish” and/or “White fish”, while AA was not associated with the consumption of any specific food items. It is noteworthy that AA is exclusively found in animal-derived foods because plants cannot synthesize C-20 LCPUFAs, including AA. In fact, compared with omnivorous women, vegetarian women had lower blood concentrations of AA during pregnancy [44]. Several decades ago, the Japanese diet consisted mainly of rice, fish, beans, and vegetables. However, recent dietary changes towards a more westernized diet, which includes more animal-derived foods, might have resulted in higher serum AA levels in breastfed infants compared to those fed CMF, even when it was supplemented with LCPUFA, including AA.

Infants predominantly fed CMF had significantly higher levels of LA in their serum than breastfed infants. Moreover, the levels of ALA did not appear to be affected by feeding patterns. The CMF used in this study was supplemented with a higher percentage of LA (15.9%) than ALA (1.6%), both of which are found in high percentages in Japanese breast milk [40]. The findings indicate that the LA content in CMF was sufficient to increase serum LA levels in infants, but the ALA content was insufficient. Since infants’ bodies can convert ALA and LA into DHA and AA, respectively [45], it was expected that infants fed CMF supplemented with ALA and LA would have higher serum DHA and AA levels. However, infants predominantly fed CMF supplemented with higher ALA and LA levels than breast milk had significantly lower levels of DHA and AA in their serum than breastfed infants. In addition, a previous RCT showed that blood DHA and AA levels were higher in infants fed CMF supplemented with higher amounts of DHA and AA than those supplemented with more ALA and LA [46]. These findings suggest that directly supplementing CMF with DHA and AA might more effectively increase their levels in infants, as compared to supplementing CMF with their precursors, ALA and LA.

One of the strengths of this study is that it used blood samples collected from 268 infants to measure 24 different types of fatty acids, despite the challenge of performing venous blood sampling in infants aged 5 to 6 months. This unique approach sets this study apart from most other studies on LCPUFA, which have relied on measuring their concentrations in breast milk.

The results of this study should be considered in light of several limitations. First, due to a lack of information on the amount of breast milk intake, evaluation of the ratio of CMF to breast milk could not be accurately assessed despite the study being prospective. Infants exclusively fed with CMF were grouped together with those fed a mixture of breast milk and CMF in the CMF >100 mL/day category. Second, the exposure to CMF might not have been adequately classified, as the exact amount of CMF and the frequency of intake were not recorded, and data on the feeding patterns for 15% (31/207) of the participants had to be substituted with their feeding patterns at 3 months of age. Further, the amount of breast milk consumed could not usually be measured. Third, there might have been residual confounding as CMF doses were not randomized. For example, mothers who exclusively breastfeed their infants might be more likely to have a healthy lifestyle, exercise habits, and high socioeconomic status. Fourth, the questionnaire survey on food intake only considered the frequency of the consumption of specific foods and did not record the amount or cooking method, both of which would affect the results. Moreover, the questionnaires utilized in this study have not undergone validation and reliability assessments for evaluating dietary intake. Fifth, the results of the interview and questionnaire survey on food patterns might have been affected by a recall bias. Sixth, the levels of DHA in infant serum might be different from those in plasma, red blood cells, or in the brain. Seventh, the study population consisted of infants at risk of atopy, and hence, generalization of our findings to healthy infants might not be appropriate. Eighth, the sample size was calculated for the ABC trial and might not be adequate for the objectives of this cross-sectional study. Therefore, associations with food items other than “Blue-White fish” might have remained undetected. Lastly, the study was conducted in the center of Tokyo, and hence, the results might not be generalizable to populations in rural areas of Japan or to the populations of other countries.

## 5. Conclusions

The present results suggest that the infants of lactating mothers who frequently consume “Blue-back fish” and/or “White fish” and are predominantly breastfed might experience a more effective increase in serum DHA levels compared to the infants of mothers who rely on DHA-supplemented CMF alone.

## Figures and Tables

**Figure 1 nutrients-15-04338-f001:**
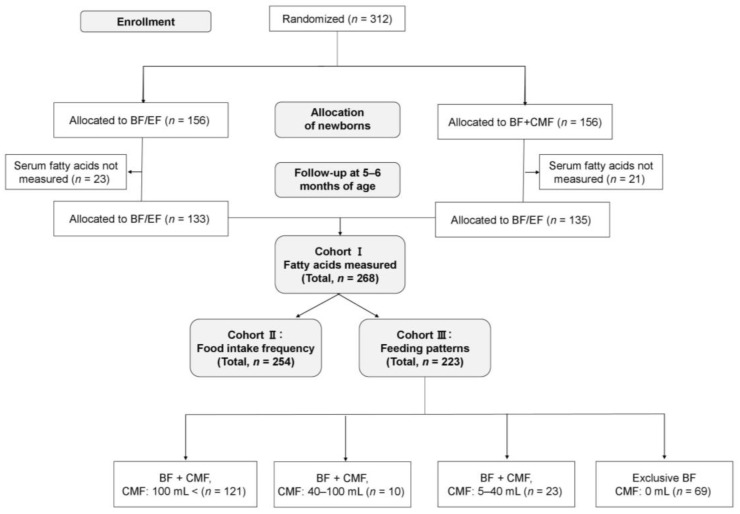
Participant flow through the ABC trial. BF + CMF, breastfeeding, and cow’s milk formula; BF/EF, breastfeeding and/or elemental formula; CMF, cow’s milk formula.

**Figure 2 nutrients-15-04338-f002:**
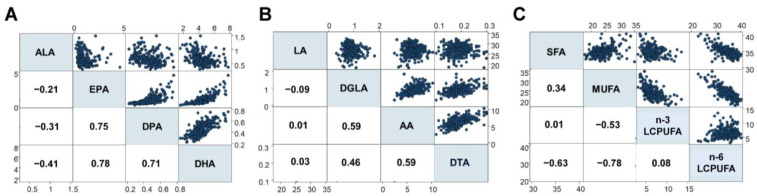
Associations between different fatty acids (%), including (**A**) n-3 LCPUFA, (**B**) n-6 LCPUFA, and (**C**) the relationship between SFA, MUFA, n-3 LCPUFA, and n-6 LCPUFA. DGLA, dihomo-γ-linolenic acid; DTA, docosatetraenoic acid; LCPUFA, long-chain polyunsaturated fatty acid. Numbers in squares represent Spearman’s rho. Blue dots represent the corresponding fatty acid levels (%).

**Figure 3 nutrients-15-04338-f003:**
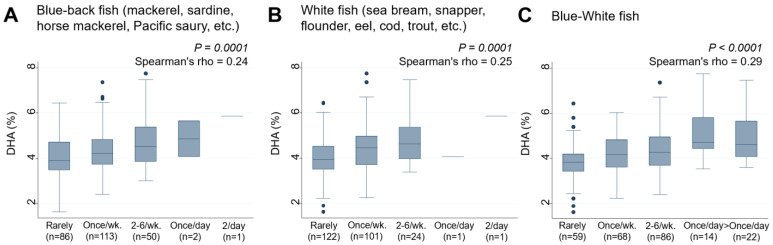
Frequency of (**A**) Blue-back fish, (**B**) White fish, and (**C**) Blue-White fish consumption by lactating mothers and the levels of serum DHA (%) in their infants. Number of non-respondents: (**A**) 2, (**B**) 5, (**C**) 5. Box presents interquartile range (IQR) from 25% to 75%. Middle line of the box means the median. The upper and lower whiskers represent minimum within 25% − 1.5*IQR and maximum within 75% + 1.5*IQR. Outliers that differ significantly from the rest of the dataset are plotted as individual points outside of the whiskers on the box-plot. A *p* value of <0.00016 was considered statistically significant.

**Figure 4 nutrients-15-04338-f004:**
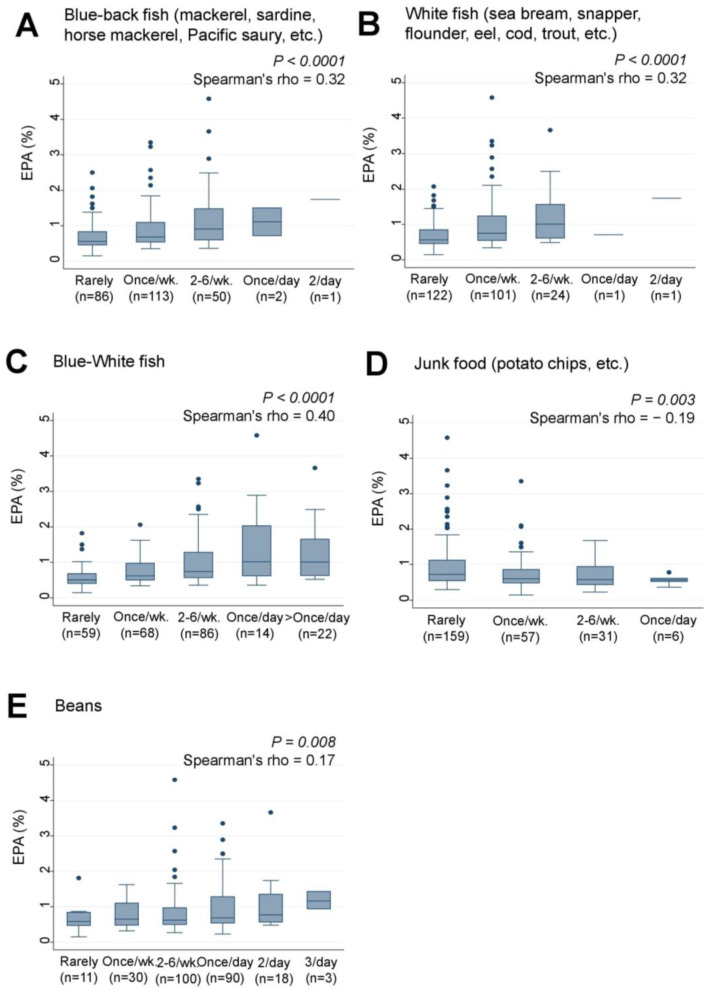
Frequency of (**A**) Blue-back fish, (**B**) White fish, (**C**) Blue-White fish, (**D**) Junk food, and (**E**) Bean consumption by lactating mothers and levels of serum EPA (%) in their infants. Number of non-respondents: (**A**) 2, (**B**) 5, (**C**) 5, (**D**) 1, (**E**) 2. Box presents interquartile range (IQR) from 25% to 75%. Middle line of the box means the median. The upper and lower whiskers represent minimum within 25% − 1.5*IQR and maximum within 75% + 1.5*IQR. Outliers that differ significantly from the rest of the dataset are plotted as individual points outside of the whiskers on the box-plot. A *p* value of <0.00016 was considered statistically significant.

**Figure 5 nutrients-15-04338-f005:**
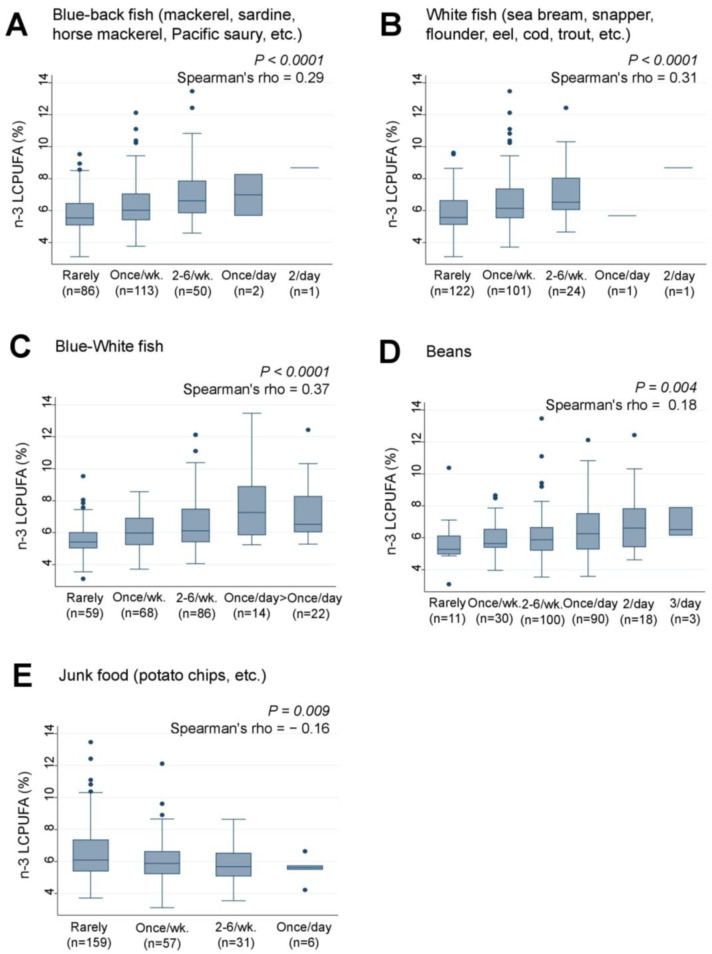
Frequency of (**A**) Blue-back fish, (**B**) White fish, (**C**) Blue-White fish, (**D**) Beans, and (**E**) Junk food consumption by lactating mothers and levels of serum n-3 LCPUFA (%) in their infants. LCPUFA, long-chain polyunsaturated fatty acid. Number of non-respondents: (**A**) 2, (**B**) 5, (**C**) 5, (**D**) 2, (**E**) 5. Box presents interquartile range (IQR) from 25% to 75%. Middle line of the box means the median. The upper and lower whiskers represent minimum within 25% − 1.5*IQR and maximum within 75% + 1.5*IQR. Outliers that differ significantly from the rest of the dataset are plotted as individual points outside of the whiskers on the box-plot. A *p* value of <0.00016 was considered statistically significant.

**Figure 6 nutrients-15-04338-f006:**
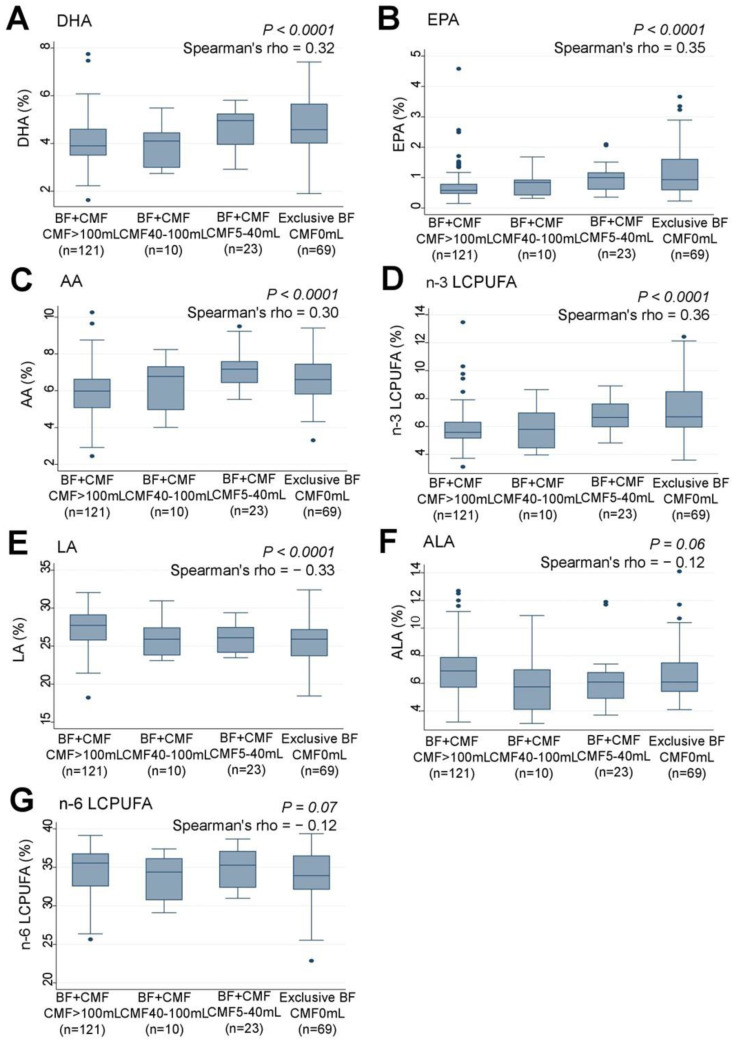
Correlations between feeding pattern grades and serum levels (%) of (**A**) DHA, (**B**) EPA, (**C**) AA, (**D**) n-3 LCPUFA, (**E**) LA, (**F**) ALA, and (**G**) n-6 LCPUFA in infants. LCPUFA, long-chain polyunsaturated fatty acid. Box presents interquartile range (IQR) from 25% to 75%. Middle line of the box means the median. The upper and lower whiskers represent minimum within 25% − 1.5*IQR and maximum within 75% + 1.5*IQR. Outliers that differ significantly from the rest of the dataset are plotted as individual points outside of the whiskers on the box-plot A *p* value of <0.00016 was considered statistically significant.

**Table 1 nutrients-15-04338-t001:** Participants’ characteristics.

	Fatty Acid Measured *n* = 268	Fatty Acid Unmeasured *n* = 42
Maternal age, mean (SD)—years	35.1 (4.3)	35.4 (4.8)
Maternal body mass index, mean (SD)—kg/m^2^	20.6 (4.3)	20.9 (2.8)
Gestational weeks, median (IQR)—weeks	39 (38–39)	39 (38–39)
Birth weight, mean (SD)—g	2993 (314)	2983 (309)
Female, no. (%)	69 (53.9)	65 (50.0)

**Table 2 nutrients-15-04338-t002:** Values of 24 types of serum fatty acids measured at 5–6 months of age (*n* = 268).

No.	Fatty Acids		μg/mL	Weight %
			Measured Value	Measured Value
			Median (IQR) Min–Max	Median (IQR) Min–Max
1	Lauric acid	12:0	25.5 (16.0–41.9) 2.9–206	0.76 (0.50–1.11) 0.11–2.96
2	Myristic acid	14:0	45.6 (31.4–67.4) 16.3–298	1.32 (1.04–1.75) 0.57–4.50
3	Myristoleic acid	14:1n-5	1.4 (1.0–2.0) 0.30–10.2	0.04 (0.03–0.05) 0.01–0.14
4	Palmitic acid	16:0	723 (654–934) 434–2280	22.3 (21.6–22.9) 19.5–26.0
5	Palmitoleic acid	16:1n-7	43.4 (32.3–55.4) 18.8–153	1.24 (1.11–1.43) 0.77–2.56
6	Stearic acid	18:0	300 (264–348) 180–846	8.67 (8.23–9.14) 6.82–10.2
7	Oleic acid	18:1n-9	719 (594–925) 327–2758	21.1 (19.5–23.1) 14.6–32.8
8	Linoleic acid	18:2n-6	931 (818–1072) 540–1840	26.9 (24.6–28.8) 18.2–31.8
9	γ-linolenic acid	18:3n-6	4.70 (3.85–5.80) 1.50–15.8	0.13 (0.11–0.16) 0.05–0.53
10	α-linolenic acid	18:3n-3	22.5 (17.0–31.4) 9.7–87.4	0.65 (0.53–0.77) 0.31–1.48
11	Arachidic acid	20:0	12.9 (11.6–14.6) 6.90–29.4	0.36 (0.34–0.40) 0.27–0.55
12	Eicosenoic acid	20:1n-9	6.60 (5.00–9.15) 2.50–28.8	0.19 (0.16–0.23) 0.10–0.48
13	Eicosadienoic acid	20:2n-6	8.90 (7.60–22.1) 4.90–22.1	0.26 (0.24–0.27) 0.18–0.35
14	5,8,11-eicosatrienoic acid	20:3n-9	2.1 (1.7–2.6) 0.7–6.1	0.06 (0.05–0.07) 0.005–0.17
15	Dihomo-γ-linolenic acid	20:3n-6	30.4 (24.3–37.1) 14.0–79.0	0.84 (0.72–1.01) 0.30–1.82
16	Arachidonic acid	20:4n-6	221 (191–257) 103–437	6.22 (5.41–7.15) 2.45–10.3
17	Eicosapentaenoic acid	20:5n-3	24.3 (17.2–39.5) 3.9–165	0.66 (0.51–1.07) 0.15–4.58
18	Behenic acid	22:0	23.1 (20.6–25.6) 13.4–41.0	0.66 (0.57–0.75) 0.35–0.99
19	Erucic acid	22:1n-9	1.6 (1.2–2.0) 0.5–4.2	0.05 (0.04–0.05) 0.005–0.09
20	Docosatetraenoic acid	22:4n-6	6.5 (5.7–7.6) 2.8–14.2	0.19 (0.16–0.21) 0.10–0.30
21	Docosapentaenoic acid	22:5n-3	16.0 (12.6–19.0) 4.6–46.1	0.44 (0.36–0.53) 0.22–0.76
22	Lignoceric acid	24:0	19.5 (17.4–22.0) 10.3–32.6	0.56 (0.48–0.65) 0.26–0.91
23	Docosahexaenoic acid	22:6n-3	150.0 (127.4–178.3) 42.8–368.6	4.17 (3.69–4.90) 1.63–7.75
24	Nervonic acid	24:1n-9	41.7 (35.0–46.8) 24.2–85.0	1.19 (0.96–1.41) 0.52–1.98

IQR, interquartile range.

**Table 3 nutrients-15-04338-t003:** Associations between frequency of maternal intake of specific foods and infant serum DHA levels.

No.	Food Item	*n*	rho	*p* Value
	**Fish**			
1	Blue-back fish: Mackerel, sardine, horse mackerel, Pacific saury (including canned)	252	0.24	0.0001
2	White fish (sea bream, snapper, flounder, eel, cod, trout, etc.)	249	0.25	0.0001
3	Salmon (including canned salmon)	250	0.03	0.64
4	Tuna (canned tuna, sashimi, etc.)	252	−0.02	0.78
5	Swordfish	249	0.01	0.82
	**Nuts**			
6	Peanuts (including peanut butter)	249	−0.06	0.36
7	Walnuts	250	−0.01	0.92
8	Almonds	251	0.04	0.49
9	Cashew nuts	250	0.06	0.32
10	Macadamia nuts	251	0.04	0.50
11	Hazelnuts	249	−0.00	0.98
12	Coconut (including that in processed foods)	244	−0.02	0.78
	**Dairy products**			
13	Cow’s milk	254	0.11	0.07
14	Cheese	252	−0.05	0.45
15	Cream (including cream for coffee)	254	−0.11	0.07
16	Ice cream	253	0.05	0.44
17	Butter	249	0.01	0.90
	**Egg**			
18	Heated eggs	252	−0.03	0.62
19	Raw eggs	221	0.02	0.79
20	Mayonnaise	251	−0.06	0.33
21	Fish eggs (salmon roe, flying fish roe, sea urchin, dried mullet roe, caviar, etc.)	250	−0.02	0.81
	**Vegetable oil**			
22	Unspecified vegetable oil	247	−0.04	0.55
23	Olive oil	250	0.03	0.61
24	Rapeseed oil	268	−0.05	0.45
25	Sesame oil	268	−0.08	0.21
26	Safflower oil	268	−0.09	0.16
27	Other	268	−0.01	0.92
28	Margarine	248	−0.07	0.25
	**Fried food**			
29	Junk food (potato chips, etc.)	253	−0.11	0.08
30	Instant noodles	250	−0.12	0.06
31	Fried food at home (tempura, fried chicken, fried potatoes, etc.)	254	0.03	0.59
32	Fried food from outside (tempura, fried chicken, fried potatoes, etc.)	254	−0.11	0.09
	**Meat**			
33	Beef	246	0.05	0.43
34	Pork	252	0.05	0.46
35	Chicken	251	0.08	0.23
36	Processed meat (sausage, salami, hotdogs, bacon, etc.)	253	−0.05	0.40
37	Hamburger at fast-food restaurant	248	−0.08	0.22
	**Beans**			
38	Soybean (natto, tofu, miso, green soybeans, soy milk) and sweet red (adzuki) beans	252	0.14	0.02

A *p* value of <0.00016 was considered statistically significant. “*n*” denotes the number of respondents to the questionnaire.

**Table 4 nutrients-15-04338-t004:** Univariate and multivariate regression analysis of factors associated with serum DHA levels in infants.

	Univariate Analysis			Multivariate Analysis					
	Model I			Model II			Model III		
	CE	95% CI	*p* Value	CE	95% CI	*p* Value	CE	95% CI	*p* Value
Blue-White fish	0.29	0.19–0.39	<0.001	0.29	0.18–0.41	<0.001	0.24	0.11–0.38	0.001
Feeding patterns	0.25	0.15–0.35	<0.001	0.18	0.09–0.28	<0.001	0.17	0.05–0.29	0.006
Allocated group in ABC trial	0.18	−0.07–0.44	0.15				0.07	−0.23–0.37	0.64
Maternal age	−0.02	−0.05–0.01	0.27				0.01	−0.02–0.04	0.54
Maternal body mass index	0.01	−0.03–0.04	0.74				0.00	−0.03–0.03	0.94
Gestational weeks	0.05	−0.01–0.11	0.13				0.05	−0.10–0.19	0.54
Birth weight	0.00	−0.00–0.00	0.20				0.31	−0.00–0.00	0.72
Female	0.19	−0.06–0.44	0.13				0.31	0.00–0.63	0.05

CE, coefficient; 95% CI, 95% confidence interval. Model I presents the results of univariate regression analysis evaluating the association of infant serum DHA levels with six variables: “Blue-White fish” intake, feeding pattern, allocation group in the ABC trial, maternal age, maternal body mass index, gestational age in weeks, birth weight, and infant sex; Model II presents the results of multivariate regression analysis, including the factors of “Blue-White fish” consumption and feeding pattern, both of which showed significant associations in Model I; and Model III presents the results of multivariate regression analysis including all six variables. A *p* value of <0.00016 was considered statistically significant.

## Data Availability

The data in the article for the clinical trials will not be made available because permission was not obtained from the study participants to share their data publicly. Other data, the code book, and analytic code will be made available on request.

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
