# Peer review of "Impact of Maternal Fish Consumption on Serum Docosahexaenoic Acid (DHA) Levels in Breastfed Infants: A Cross-Sectional Study of a Randomized Clinical Trial in Japan"

_nutrients, 2023, doi:10.3390/nu15204338_

Round 1

Reviewer 1 Report

Thank you for trusting me review this manuscript.

This was a cross-sectional study of a randomized clinical trial in Japan. Authors used serum samples from 268 infants participating the RCT trail.

Here goes my comments:

(1)   It is recommended that, as a RCT trail, sample size estimation and randomization details shall be described in the method section of the manuscript.

(2)   The ABC trial included 312 pregnant women and these women were randomly assigned to…” (lines 131-132) cause me a bit confused in how 100% of the 312 pregnant women participated in the study after their deliveries. As usually, there must be a certain percentage of loss of follow-up.

(3)   If there is no gender difference in the results, it is recommended to add relevant results and data analysis in the result section, as well as in the discussion section.

Reviewer 2 Report

The author still needs to address some issues.

1. The author should annotate the table in the discount results. For example, the annotations in the table should indicate which variables are included in the model.

2. Has the author tested the level of DHA in maternal blood samples and evaluated the relationship between DHA levels in maternal circulation and neonatal circulation.

3. This study is a post hoc study. Has the author fully considered whether such a small sample size is sufficient to support the current conclusion.

4. The author used fewer dietary questionnaire entries compared to other studies. Has this dietary questionnaire been evaluated for reliability and validity?

Moderate editing of English language required

Reviewer 3 Report

A review of the article "Effect of maternal fish consumption on serum docosahexaenoic acid (DHA) levels in breastfed infants: a cross-sectional study of a randomized clinical trial in Japan" by Ayu Kasamatsu, Hiroshi Tachimoto and Mitsuyoshi Urashima.

The article is a valuable contribution to research on the effects of maternal nutrition on infant health, with a special focus on the role of docosahexaenoic acid (DHA) in infant brain development. The authors conducted a cross-sectional study on a sample of 268 infants participating in a randomized clinical trial in Japan, where fish consumption is common. The study analyzed maternal dietary habits, infant feeding patterns and serum levels of DHA and other long-chain polyunsaturated fatty acids (LCPUFAs) in infants at 5-6 months of age.

The results of the study are interesting and suggest a significant effect of maternal fish consumption on infants' serum DHA levels. The authors showed positive associations between mothers' consumption of "Bluefish" and "White Fish" and higher serum DHA levels in infants. In addition, infants who were primarily breastfed had significantly higher serum DHA levels compared to those who received higher amounts of DHA-enriched cow's milk.

An interesting aspect of this study is the inclusion of other factors, such as "blue-white fish" and "feeding patterns," which remained significantly and independently associated with serum DHA levels. This suggests that maternal dietary diversity and the priority of breastfeeding over DHA-enriched cow's milk may synergistically affect DHA levels in infants.

This article makes an important contribution to understanding the role of maternal nutrition in shaping infant health, particularly in the context of brain development. It points to potential strategies for improving DHA levels in infants, which may contribute to healthier brain development and function. However, it is worth noting that this study is cross-sectional, meaning that causality cannot be inferred, only existing relationships. Therefore, further research and intervention studies may be needed to confirm these findings.

The introduction provides a solid foundation for the study, but it could be further improved by clarifying and expanding a few key points:

A clear statement of purpose: Although the introduction mentions the purpose of the study, it could be made more specific. A clear, concise statement of the study's purpose should be included to inform the reader of the main purpose of the study.

Background information: It would be helpful to provide a little more background information on the importance of DHA for infant brain development and the existing literature on the subject. This will help set the context and justify the study.

Formatting of citations: Sources in the text should be consistently cited using a specific citation style (e.g., APA or Chicago). This will improve the overall readability and professionalism of the text.

Flow and consistency: The text could be organized in a way that provides better flow and consistency. For example, grouping information related to DHA and its meaning together would make it easier for the reader to follow the logic.

Clarity about the study design: Although it was mentioned that this is a cross-sectional study, it may be helpful to briefly explain what a cross-sectional study is to readers who may not be familiar with the term.

Rationale for choosing Japan as the study site: The rationale for choosing Japan as the study site was briefly mentioned, but could be expanded upon. Why is Japan an ideal location? Are there specific cultural or dietary factors unique to Japan that make it relevant to this study?

Study population: Providing some information about the study population, such as the number of mothers and infants, would give readers a sense of the scale of the study.

Identifying the research gap: Clearly identifying the research gap that this study aims to address would make the significance of the study clearer.

1. study design

- Introduce a short, clear definition of the study so that the reader immediately understands what is being discussed.

- Provide more information about recruitment, such as the number of participants and how they were selected.

- Consider better highlighting the study in a scientific or practical context.

2.2 Population of the study

- Explain why specific inclusion and exclusion criteria were chosen for the study. This may help the reader understand what population groups were studied.

- Consider giving examples of atopic diseases so that the reader can better understand what includes atopic risk.

2.3 Infant formula and intervention

- Provide more detailed information on cow's milk formula itself (CMF) and amino acid formula (EF). You can provide information on the nutritional composition, vitamins, minerals, etc.

- Explain why these two formulas were chosen and the advantages of the choice.

- Briefly and clearly describe which groups received which types of formula and why.

Overall:

- Make sure that the entire text is consistent in terms of style and terminology.

- Make sure the text is clear, avoid unnecessary abbreviations and terms.

- You can provide more information about the potential benefits and importance of the study to the scientific community or patients.

- Adapt the literature according to MDPI styles. 

Greetings and congratulations!

Reviewer 4 Report

The authors carried out an extensive study in that they investigated that relationships among a) maternal diet, b) infant feeding practices and c) fatty acid composition of serum lipids at the age of 5-6 months. I cannot really tell whether the data obtained in the study might be worthy. However, I can tell with reasonably certainty that the way of description of methods and data presentation in the present paper is unacceptable.

First, I would like to call the authors attention that data obtained in a study are usually to be published within one single scientific paper. To write that “The associations between serum LCPUFA levels and the risk of atopy will be provided separately” (lines 63 and 64) suggests intentional compartmentalisation of data, which is not to be encouraged. But this is by far not the biggest problem with the present manuscript.

The major problems are with the presentation of ungrounded, therefore scientifically meaningless data throughout the paper. In Table 1 the authors present “standard values” for serum fatty acid measurements. These standard vales are presumably the reference values of the Standard reference Laboratory (SRL) Inc. in Hachioji, Tokyo, Japan (line 110 and 111). Whose serum values are these (adults, or children, or expecting women???), how many measurements do they represent, when and how were these values obtained: no information is given.

Similarly, the authors just declare that “Data on maternal age, body mass index, gestational weeks, birth weight, neonatal sex, and other variables were obtained from the neonatal summary record at births” (lines 92 to 95), but no actual data are shown. It remains totally undefined whether the mothers were younger or older, they had their first or not the first baby, were obese or not, etc. We know nothing about the study population. Nevertheless, these undefined data are entered into regression analysis (Table 3). The numbers in Table 3 show mostly coefficients among items within a black box.

Similarly, associations between frequency of maternal intake of specific foods and infant serum DHA levels are show for 38 food items in Table 2, whereas actual distribution of frequencies are provided only for 5 food items (Figures 3 to 5). However, the picture that can be deciphered form the figures is very much disturbing. Just how comes that only one mother consumed blue-back fish twice per day, one mother consumed white fish twice per day, but not less than 22 mothers consumed blue-white fish twice per day (Fig 3). One should ask the question just how precise these food frequency estimates are? When and where were the questionnaires validated?

Minor editing of English language required. 

Round 2

Reviewer 2 Report

Accept in present form

Minor editing of English language required

Reviewer 4 Report

The MS improved considerably during the review process. My recommendations and the suggestions of the following minor modifications: 1. Table S4 contains the data of the subjects investigated. Normally such table should be inserted into the main text. Why not insert this small table before present Table 1 and renumber the tables? 2. Figures 3 to 6 are inserted in an unnecessary large format into the MS. I strongly recommend to lessen the size of these figures, e.g. to compress them to a "one figure - one page" format, at least.
